# Initiation of cytosolic plant purine nucleotide catabolism involves a monospecific xanthosine monophosphate phosphatase

Katharina J. Heinemann[1,3], Sun-Young Yang[2,3], Henryk Straube [1], Nieves Medina-Escobar[1], Marina Varbanova-Herde[1], Marco Herde[1], Sangkee Rhee [2✉] & Claus-Peter Witte [1✉]

In plants, guanosine monophosphate (GMP) is synthesized from adenosine monophosphate via inosine monophosphate and xanthosine monophosphate (XMP) in the cytosol. It has been shown recently that the catabolic route for adenylate-derived nucleotides bifurcates at XMP from this biosynthetic route. Dephosphorylation of XMP and GMP by as yet unknown phosphatases can initiate cytosolic purine nucleotide catabolism. Here we show that *Arabidopsis thaliana* possesses a highly XMP-specific phosphatase (XMPP) which is conserved in vascular plants. We demonstrate that XMPP catalyzes the irreversible entry reaction of adenylate-derived nucleotides into purine nucleotide catabolism in vivo, whereas the guanylates enter catabolism via an unidentified GMP phosphatase and guanosine deaminase which are important to maintain purine nucleotide homeostasis. We also present a crystal structure and mutational analysis of XMPP providing a rationale for its exceptionally high substrate specificity, which is likely required for the efficient catalysis of the very small XMP pool in vivo.

[1] Leibniz Universität Hannover, Department of Molecular Nutrition and Biochemistry of Plants, Herrenhäuser Strasse 2, 30419 Hannover, Germany. [2] Seoul National University, Department of Agricultural Biotechnology, 151-921 Seoul, Republic of Korea. [3]These authors contributed equally: Katharina J. Heinemann, Sun-Young Yang. ✉email: srheesnu@snu.ac.kr; cpwitte@pflern.uni-hannover.de

Purine metabolism in plants and other eukaryotes differs in several aspects. For example, in plants, purine nucleotide biosynthesis generating AMP is localized in chloroplasts, whereas in mammals or in yeasts it is located in the cytosol. However, plant GMP biosynthesis, starting from AMP, occurs in the cytosol (Fig. 1a and Supplementary Fig. 1)[1,2]. In contrast to mammals, plants are able to degrade purine nucleotides completely, disintegrating the purine ring via intermediates like urate and allantoate into glyoxylate, carbon dioxide and ammonium[3,4]. Recently, it was shown in *Arabidopsis thaliana* that there are two independent entry reactions into purine nucleotide catabolism[5]. One is leading from xanthosine monophosphate (XMP) to xanthosine presumably catalyzed by an XMP phosphatase (XMPP). This reaction may serve to degrade the adenylate-derived purine nucleotides. The other also generates xanthosine but via the dephosphorylation of GMP to guanosine by a so far unknown

**Fig. 1 Scheme of GMP biosynthesis and purine nucleotide catabolism and phylogenetic analysis of SDTL proteins. a** GMP biosynthesis and purine nucleotide catabolism in the cytosol are initiated from AMP, which is deaminated to inosine monophosphate (IMP) by AMP deaminase (AMPD). IMP is oxidized by IMP dehydrogenase (IMPDH) to xanthosine monophosphate (XMP). For biosynthesis of GMP, XMP is aminated by GMP synthetase (GMPS). GMP and XMP can be dephosphorylated by XMP phosphatase (XMPP) and GMP phosphatase (GMPP) yielding guanosine and xanthosine, respectively. Both enzymes have not yet been described. Guanosine is deaminated to xanthosine by guanosine deaminase (GSDA). In plants, xanthosine can only be degraded and not be salvaged into nucleotides. By contrast, guanosine can be salvaged to GMP by a so far unknown guanosine kinase (GK) and guanine, which is not a purine nucleotide catabolism intermediate in Arabidopsis, can be salvaged by hypoxanthine guanine phosphoribosyltransferase (HGPRT). Xanthosine is hydrolyzed to ribose and xanthine by the nucleoside hydrolase heterocomplex (NSH1/NSH2) which is not functional in the absence of NSH1. Xanthine is oxidized by xanthine dehydrogenase (XDH) to urate. Via the intermediates (*S*)-allantoin and allantoate, urate can be fully degraded to glyoxylate, carbon dioxide, and ammonium. The gene locus identifiers are given where known. A scheme including chemical formulas is shown in Supplemental Fig. 1. **b** Maximum Likelihood tree constructed with SDTL sequences from 15 phylogenetically distant vascular plant and three moss species. The tree with the highest log likelihood is shown. In total, 1000 bootstraps were performed and only bootstrap values over 70% are shown. Species names and accession numbers are given in Supplementary Table 1. Supplementary Data 1 contains the multiple protein alignment from which the tree was calculated.

GMP phosphatase (GMPP) and the subsequent deamination of guanosine by guanosine deaminase (GSDA), an enzyme found only in plants (Fig. 1a and Supplementary Fig. 1)[6]. Because xanthosine cannot be converted into other nucleosides and cannot be re-phosphorylated (salvaged) to XMP in Arabidopsis[7,8], the XMPP and GSDA reactions both lead irreversibly into purine catabolism.

Although several enzymes with nucleotide monophosphate (NMP) phosphatase activity have been described in eukaryotes, there are only few examples where their physiological role could be clearly demonstrated[9,10]. Why is the functional assignment of NMP phosphatases so difficult? Possible reasons are (i) that NMPs are central metabolites and their pool sizes are affected by a multitude of reactions so that single mutations hardly change the pools at all; (ii) eukaryotes have many enzymes with NMP phosphatase activity often with a broad substrate spectrum complicating the assignment of specific physiological roles; (iii) NMP phosphatases may act partially redundant in vivo.

In this work, we were able to identify a plant-specific NMP phosphatase, the XMP phosphatase. We have characterized the enzyme biochemically and report its crystal structure with bound substrate elucidating its specificity determinants. Most importantly, we demonstrate that XMPP is involved in purine nucleotide catabolism in vivo.

## Results and discussion

**Identification and molecular characterization of XMPP**. In *Saccharomyces cerevisiae* two homologous enzymes, Sdt1p (Suppressor of disruption of *TFIIS*) and Phm8p (Phosphate metabolism protein 8), were shown to hydrolyze NMPs in vitro. Phm8p but not Sdt1p is required in vivo for purine and pyrimidine mononucleotide degradation[10].

Because Phm8p also shows activity with XMP and GMP, we searched for genes in the Arabidopsis genome encoding homologous proteins to Phm8p/Sdt1p with the aim to identify XMP- and GMP-specific plant phosphatases (Fig. 1a and Supplementary Fig. 1). Five proteins homologous to Phm8p/Sdt1p were found which we called SDTL for Sdt1p-like as they were more similar to Sdt1p than to Phm8p. Because enzymes of primary metabolism are usually highly conserved in plants, we assumed that only SDTLs that are present in all vascular plants are good XMP/GMP phosphatase candidates. To identify such SDTLs, a maximum likelihood tree was constructed using the full complement of SDTL sequences from 15 phylogenetically distant vascular plants and 3 moss species (Fig. 1b and Supplementary Table 1). Four major clades of SDTL proteins were found. The clades SDTL1, SDTL2, and SDTL3 each contain at least one representative sequence from every analyzed vascular plant species, whereas the SDTL4 clade lacks sequences from several plant species, in particular from the monocots. Each of the mosses only has a single SDTL not associated with any of the clades. The clades likely represent groups of orthologous proteins with identical or at least similar functions. The SDTL1 clade is more distant from the others and contains only a single protein from each of the analyzed vascular plants, whereas in other clades also sub-clades are found indicating some functional diversification.

We transiently expressed cDNAs coding for C-terminal Strep-tagged variants of the SDTL proteins from the clades 1, 2, and 3 of Arabidopsis in leaves of *Nicotiana benthamiana*, affinity purified the proteins, and screened their activity with XMP and GMP as substrates. GMP phosphatase activity was not observed for any of the enzymes but SDTL from clade 1 showed activity with XMP. We extended the substrate survey for this enzyme to other mononucleotides and found that it is highly specific for

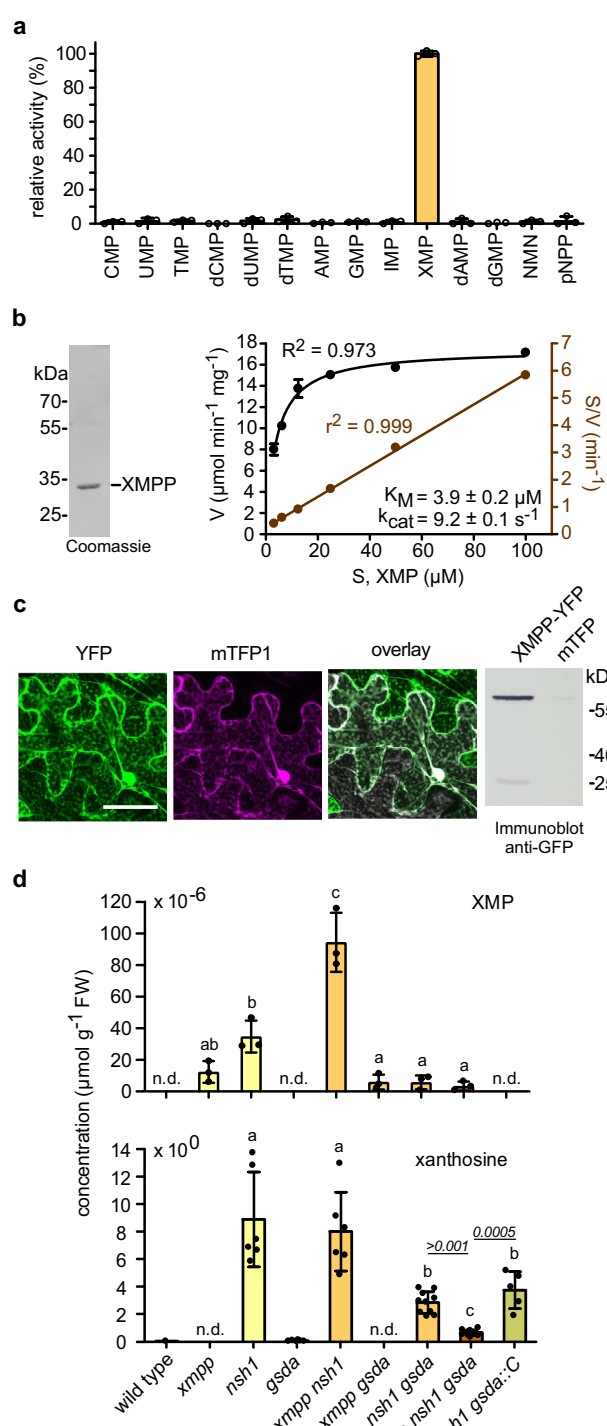

XMP (Fig. 2a). Such high specificity is unusual for NMP phosphatases. The enzyme has a $K_M$ value of $3.9 \pm 0.2\,\mu M$ and a $k_{cat}$ value of $9.2 \pm 0.1\,s^{-1}$ for XMP (Fig. 2b). An N-terminal Strep-tagged variant has similar kinetic constants (Supplementary Fig. 2) indicating that the tags and the tag position do not interfere with the activity. Based on these data, we named the enzyme XMP phosphatase (XMPP). The subcellular localization of XMPP was investigated by confocal laser scanning microscopy in the leaves of *N. benthamiana* transiently expressing the enzyme as C-terminal yellow fluorescent protein (YFP)-tagged variant. XMPP is present in the cytosol and the nucleus (Fig. 2c) as are

**Fig. 2 Characterization of the XMPP protein and of *XMPP* genetic variants in the context of other mutants of purine catabolism genes in seeds. a** Relative enzymatic activity of XMPP-HAStrep with 100 μM (deoxy)mononucleotides including nicotinamide mononucleotide (NMN) and the general phosphatase substrate *para*-nitrophenyl phosphate (pNPP). Error bars are SD, $n = 3$. An XMP conversion rate of $12.1 \pm 0.2$ μmol mg$^{-1}$ min$^{-1}$ was set to 100%. **b** XMPP-HAStrep affinity purified from leaf extracts of *Nicotiana benthamiana* after transient expression. Left panel, Coomassie-stained SDS-gel with the purified enzyme. Right panel, determination of the kinetic constants at 22 °C with the data fitted according to Michaelis Menten (left axis) or Hanes (right axis). Error bars are SD, $n = 3$. **c** Confocal fluorescence microscopy images of a lower epidermis cell of *N. benthamiana* co-expressing XMPP-YFP and the cytosolic cyan fluorescent protein mTFP1. From left to right, YFP channel, mTFP1 channel, overlay of both channels. Scale bar = 50 μm. Immunoblot of *N. benthamiana* leaf extracts expressing either XMPP-YFP or free mTFP1 probed with anti-GFP antibody demonstrating the high stability of the fusion protein. The immunoblot was repeated twice with similar results. **d** Quantification of XMP and xanthosine in seed extracts of the indicated genotypes. The *xmpp-1* allele (Supplementary Fig. 3a, b) was used in all instances. The complementation line (*xmpp nsh1 gsda::C*) contains a transgene encoding an N-terminal TwinStrep-tagged XMPP expressed from a native promoter (Supplementary Fig. 3c, d). Error bars are SD, $n = 3$ for XMP analysis and for xanthosine analysis $n = 6$ but for *xmpp nsh1 gsda::C* $n = 5$ and for *nsh1 gsda* and *xmpp nsh1 gsda* $n = 10$. A repeat ($n$) is a seed sample from an independent mother plant grown in parallel with all other plants of the experiment. Statistical analysis with two-sided Tukey's pairwise comparisons using the sandwich variance estimator. Different letters indicate $p$ values < 0.05. Some $p$ values are indicated above the columns in italic numbers; all $p$ values can be found in the Source Data file.

other enzymes upstream and downstream of XMPP, i.e. IMP dehydrogenase (IMPDH), GMP synthetase (GMPS)[2], guanosine deaminase (GSDA)[6], and the nucleoside hydrolase heterocomplex (NSH1/NSH2)[11–13] (Fig. 1a).

**The function of XMPP in purine nucleotide catabolism in vivo.** To investigate whether XMPP is involved in degrading XMP in vivo, we used a series of single (*xmpp, nsh1, gsda*), double (*xmpp nsh1, xmpp gsda, nsh1 gsda*), and triple (*xmpp nsh1 gsda*) mutants as well as two complementation lines containing a *TwinStrep-XMPP* transgene driven from the native promoter (Supplementary Fig. 3). The pools of XMP, xanthosine, guanosine, and guanine in these genetic variants were quantified in fresh seeds, because it is known that purine catabolism is active during seed development. The XMP concentration was close to the detection limit in most genotypes but was slightly elevated in the *xmpp* line (Fig. 2d). In *nsh1* seeds, more XMP was detected and in the *xmpp nsh1* seeds the largest XMP pool was observed, suggesting that XMPP is involved in XMP catalysis in vivo. The absence of NSH1, which plays a central role for xanthosine hydrolysis[5], has a positive effect on the XMP concentration possibly because the strong accumulation of xanthosine in *nsh1* seeds[5,11,13] (Fig. 2d) leads to a partial inhibition of the two XMP converting enzymes: XMPP and GMPS (Fig. 1a). For XMPP, we confirmed an inhibition by xanthosine (Supplementary Fig. 4). XMP accumulation in *xmpp* seeds is comparatively low probably because XMP is easily aminated by GMPS to GMP, which can be degraded by an unknown GMP phosphatase (GMPP) and GSDA via guanosine to xanthosine (Fig. 1a). This hypothesis is supported by hyperaccumulation of guanosine and guanine in seeds with *xmpp gsda* background compared to *gsda* seeds (Supplementary Fig. 5). GMP can also be phosphorylated leading in tendency to GTP hyperaccumulation in this double mutant

background, which is prevented and even over-compensated by the introduction of an XMPP transgene (Supplementary Fig. 6). It has been noted before that guanosine accumulation in *gsda* plants results in higher concentrations of guanylates and in consequence adenylates[14]. The GMPP and GSDA reactions seem to guard guanylate homeostasis.

Xanthosine is the first common product of direct XMP dephosphorylation by XMPP and of GMP degradation by GMPP and GSDA (Fig. 1a). The xanthosine pool size in *nsh1* background may therefore serve as a proxy for assessing if both routes are operative. Whereas in *nsh1 gsda* seeds the xanthosine concentration is reduced strongly compared to the *nsh1* background, this is not the case in *xmpp nsh1* seeds (Fig. 2d), suggesting at first sight that the XMPP route plays no role. However, one needs to bear in mind that XMP can easily be converted to GMP by GMPS, especially if XMPP is not functional, thus only in *xmpp nsh1 gsda* seeds one can assess whether XMPP contributes to the xanthosine pool in vivo. This is indeed the case because in the triple mutant the xanthosine concentration is strongly reduced and this effect is reversed if the triple mutant expresses an *XMPP* transgene. These data demonstrate that XMPP catalyzes the dephosphorylation of XMP in seeds representing an entry point of the adenylate-derived nucleotides into purine catabolism in vivo. The XMPP reaction is likely the main entry point for the adenylates into catabolism, whereas the GMPP reaction serves for the degradation of the guanylates. Note, that xanthosine cannot be salvaged in Arabidopsis whereas guanosine can[8]—thus the XMPP reaction leads irreversibly into purine nucleotide degradation.

Purine catabolism is enhanced by prolonged darkness[5,15]. Therefore, we assessed the role of XMPP in seedlings exposed to a night prolonged by 48 h in comparison to seedlings in a 16 h day/ 8 h night regime. Dark treatment led to a marked increase of the steady-state pool sizes of urate and allantoate (Fig. 3a), which are intermediates of purine nucleoside catabolism[4,16] (Fig. 1a and Supplementary Fig. 1). In *gsda* seedlings under long-day conditions, the urate pool was already smaller than in the wild type or in *xmpp* seedlings, while in *xmpp gsda* plants, the urate concentration was even more reduced and allantoate could not be detected anymore. In the prolonged night, urate and allantoate concentrations rose but not in the *xmpp gsda* seedlings, demonstrating that the degradation of XMP by XMPP and of GMP via GSDA feeds these pools. The dark-exposure experiment was repeated including a line expressing an *XMPP* transgene in *xmpp gsda* background (Supplementary Fig. 3c), which complemented metabolic changes observed in the *xmpp gsda* line compared with the *gsda* line (Supplementary Fig. 7). Together these results show that XMP and GMP dephosphorylation can initiate cytosolic purine nucleotide catabolism, but it is not possible to quantify the relative contributions of XMP versus GMP degradation from the data. Reasons are that the XMPP reaction can be bypassed in *xmpp* background via GMP and guanosine (Fig. 1a), and that in *gsda* seedlings guanosine accumulates so strongly that it partially inhibits the NSH1/ NSH2 complex, which leads to xanthosine buildup in the dark (Fig. 3a and Supplementary Fig. 7), and consequently to partial inhibition of XMPP (Supplementary Fig. 4). The dark-induced xanthosine accumulation in *gsda* background is not observed in *xmpp gsda* seedlings, again showing that the XMPP reaction directly contributes to the xanthosine pool in vivo. Because XMP cannot be channeled into degradation in *xmpp gsda* plants, these double mutants accumulate more guanosine than *gsda* seedlings in the dark (Fig. 3a). In an independent experiment we observed that the *xmpp gsda* plants also over-accumulate GTP and in consequence ATP in tendency, which is prevented by the introduction of an *XMPP* transgene (Fig. 3b and Supplementary

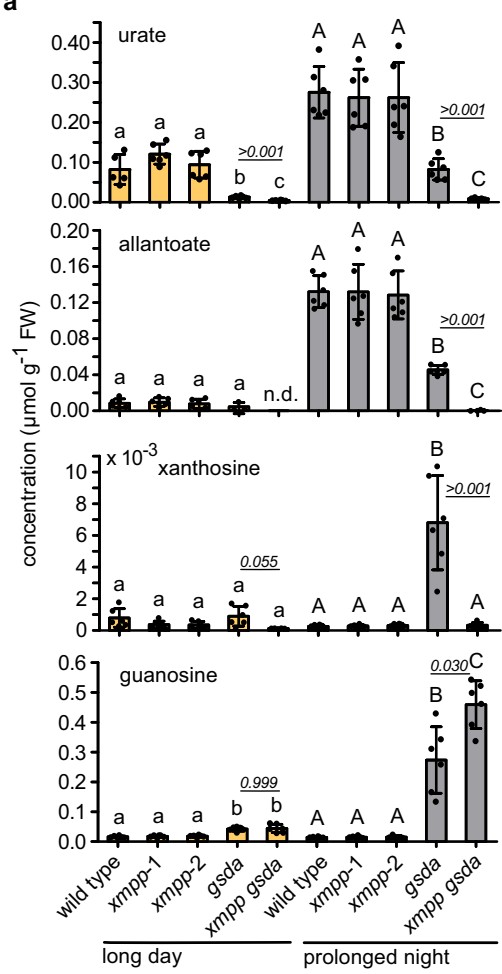

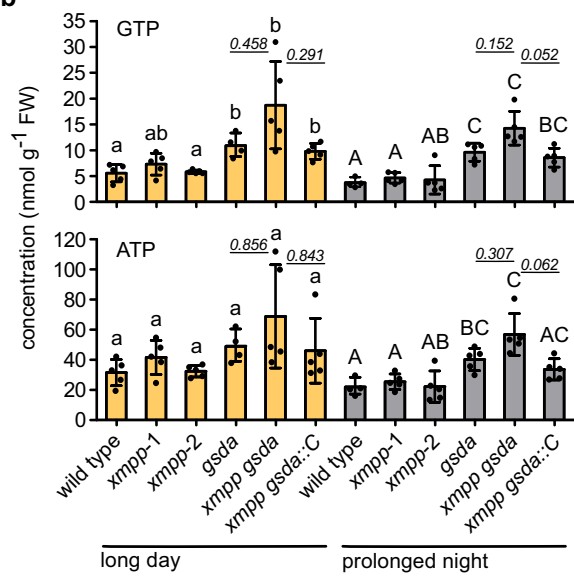

**Fig. 3 Characterization of metabolic alterations in *XMPP* genetic variants in the context of other mutants of purine catabolism genes in seedlings under long-day or prolonged-night conditions.** Two-sided Tukey's pairwise comparisons using the sandwich variance estimator were employed for statistical analyses. Different letters indicate *p* values < 0.05. Samples from long day and prolonged darkness were independently analyzed indicated by small and capital letters. Some *p* values are indicated above the columns in italic numbers, all *p* values can be found in the Source Data file. **a** Urate, allantoate, xanthosine, and guanosine pool sizes in the indicated genotypes in seedlings. The seedlings were grown in a constant 16 h day/8 h night regime up to day 5 after germination. At the end of the night of day 5, they were either grown the same way for another 48 h (orange bars) or exposed to darkness for 48 h (dark bars). All genotypes were grown on the same plate for one biological replicate (*n*). Error bars are SD, *n* = 6. **b** GTP and ATP pool sizes of seedlings from an independent experiment grown as described in **a**. Error bars are SD, *n* = 5 but for *gsda* in the light and the wild type in the dark *n* = 4.

The mutation of *XMPP* does not create an obvious macroscopic phenotype at any developmental stage (Supplementary Fig. 9) and also not during dark stress. This is not surprising because (1) many purine catabolism mutants of Arabidopsis do not show phenotypes under standard growth conditions except those that accumulate toxic intermediates like guanosine[15] or uric acid[16] and because (2) it is possible to bypass the XMPP reaction via GMP and guanosine as shown above. However, this bypass is energetically costly, because the GMPS reaction requires the conversion of ATP to AMP and uses glutamine for the amination of XMP. Subsequently, this amino group is released as ammonia by GSDA and must be re-assimilated into glutamine which is energized by the conversion of ATP to ADP. Thus in total the bypass requires the hydrolysis of three phosphoanhydride bonds that are spared when the XMPP reaction is used.

**The molecular basis of substrate recognition by XMPP.** XMPP is highly substrate specific (Fig. 2a). To elucidate the molecular basis for this specificity, we determined the crystal structure of XMPP with and without XMP bound (Fig. 4, Supplementary Fig. 10a and Supplementary Table 2). Crystals could only be obtained for a shortened XMPP variant lacking 13 amino acids at the C-terminus (Met1 to Ser250). Consistent with the monomeric state of the enzyme in solution (Supplementary Fig. 10b), crystalized XMPP is monomeric with a two-domain architecture (Fig. 4a). In the α/β-folded central domain with five repeating β-α units, the central parallel six-stranded β-sheet is flanked by five α-helices on both sides. A 20-residue-long loop (Pro142 to Ser161) protrudes from the β3–α8 unit, and a cap domain (Leu20 to Ser89) with a four α-helices bundle-like architecture is inserted in the β1–α5 unit. The loop probably stabilizes the orientation of the cap domain (Supplementary Fig. 11). XMPP is a member of the Haloacid Dehalogenase (HAD) protein superfamily with unique characteristics since HAD enzymes typically have either a protruding loop or a cap structure[17], but not both, like XMPP.

At the junction of the cap and the central domains, a $Mg^{2+}$ ion coordinates six ligands within 2.1 Å in a square bi-pyramidal geometry (Fig. 4b) involving Asp12 and Asp14. These aspartates are conserved (Supplementary Fig. 10a) and important for the catalytic activity of HAD enzymes. In the complex with XMP, the 5′-monophosphate moiety of XMP provides an equatorial ligand across Asp184 in the $Mg^{2+}$-coordination shell, and the ribose and xanthine moieties face toward the cap domain (Fig. 4c, d and Supplementary Fig. 12). Unlike the ribose moiety, xanthine embedded on the concave side of the cap domain interacts extensively with the enzyme via (i) hydrophobic interactions with

Fig. 8). These tendencies were repeatedly observed in the long day and in the prolonged night. It is clear that GSDA, probably in concert with the unknown GMPP, is necessary to keep guanylate and adenylate homeostasis, and that XMPP prevents adenylate-derived nucleotides destined for degradation to spill into the guanylate pool.

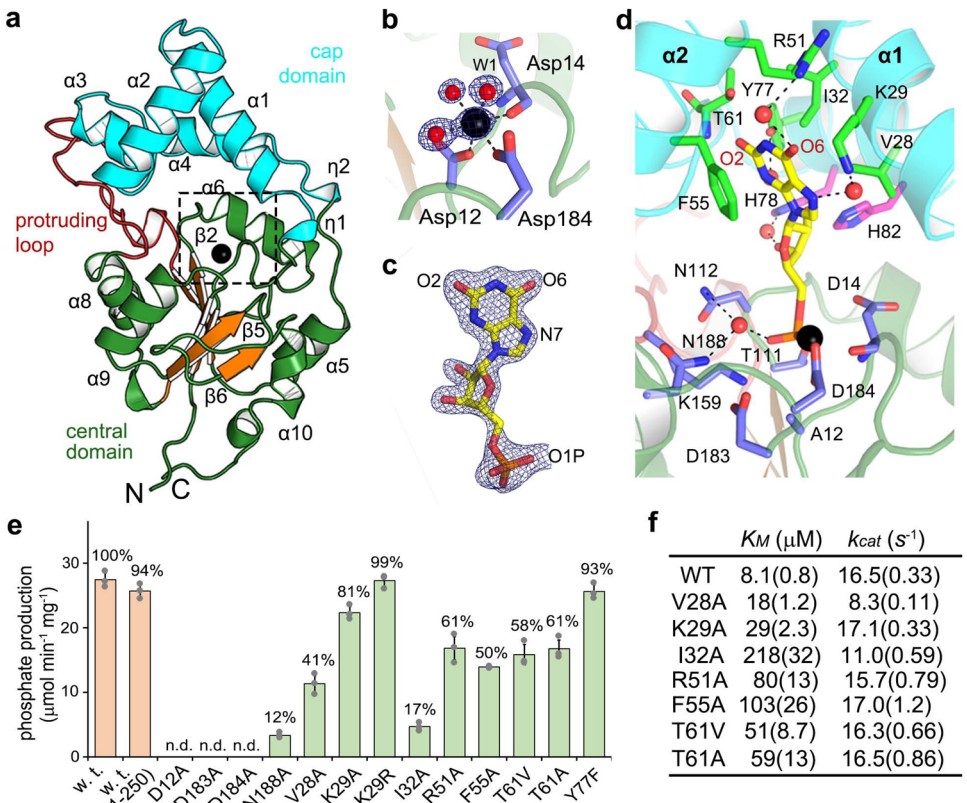

**Fig. 4 Structural and mutational analysis of XMPP. a** Structure of the unliganded XMPP. The repeating β–α units (orange, green) in the central domain include β1–α5, β2–α6, β3–α8, β4–α9, and β6–α10. Metal-binding site, dashed box. $Mg^{2+}$ ion, black sphere; cap domain, cyan; protruding loop, red. **b** A zoomed-in view of the proposed $Mg^{2+}$-binding site. Dashed lines indicate the six coordination bonds. Asp12 and water W1, axial ligands. Asp184, the main chain carbonyl oxygen of Asp14 and two water, equatorial ligands. Water (red) and metal ion (black) are overlaid with a $2Fo-Fc$ electron density map contoured at $1.0\sigma$ and $3.0\sigma$, respectively. **c** XMP as oriented in the structure of the complex overlaid with an $Fo-Fc$ electron density map contoured at $3.5\sigma$. **d** An enlarged view of the XMP-binding site oriented as in **a**. The complex with XMP was obtained with an inactive mutant, in which the putative catalytic Asp12 was replaced with alanine. Blue, residues interacting with phosphate and $Mg^{2+}$; magenta, ribose moiety-interacting residues; green, xanthine-binding residues. All residues are within 4.0 Å of XMP. Water-mediated hydrogen bonds, dashed lines. The phosphate moiety interacts with residues that are highly conserved in the HAD superfamily, including Asp14, Thr111, Asn112, Lys159, and Asn188. **e** Specific activities of XMPP variants with 125 μM XMP as substrate. Full-length enzymes with distinct point mutations expressed in *Escherichia coli* and purified via a C-terminal His tag were used. Relative activities compared to the wild-type enzyme (27.4 nmol min$^{-1}$μg$^{-1}$ set to 100%) are given. Error bars are SD, $n = 3$ repeated assays using the same enzyme preparation. n.d. for not detected. Note that the C-terminally truncated XMPP (residues 1–250) used for structure determination exhibited an almost equal specific activity to that of the full-length wild-type enzyme suggesting that the C-terminal region beyond Ser250 is not crucial for in vitro phosphatase activity. **f** Kinetic constants of various mutants of xanthine-interacting residues. SE is indicated in a parenthesis.

Val28 and Ile32 and (ii) by base stacking with Phe55, and (iii) via direct or water-mediated indirect hydrogen bonds of $O_2$ with Thr61, O6 indirectly with Arg51, and N3 and N7 indirectly with Tyr77 and Lys29, respectively (Fig. 4d).

To validate the structure-based functional assignments, enzymatic activities and kinetic constants of XMPP variants were determined. The specific activities of the XMPP variants closely reflect the assigned functional roles of the respective residues (Fig. 4d, e). $Mg^{2+}$-coordination by Asp12, Asp183, and Asp184 is linked to catalysis and is of critical importance since the respective mutants are inactive. The interaction of Asn188 with the phosphate and of various residues with the xanthine moiety are required for full activity (Fig. 4e). Further kinetic analysis revealed that mutations of xanthine-interacting residues do not affect the $k_{cat}$ but the $K_M$ values for XMP (Fig. 4f and Supplementary Fig. 13). For variants altered in hydrophobic residues, the $K_M$ increases 2.2- to 26.9-fold. For those changed in hydrogen bond-forming residues, the $K_M$ increase ranges from 3.6- to 9.9-fold. The XMP specificity is thus collectively mediated by the hydrophobic residues Val28, Ile31, and Phe55 that shape

the xanthine-binding pocket in the cap domain and by direct and indirect hydrogen bonds to Thr61, Lys29, and Arg51. These amino acids are largely conserved in putative XMPPs from other plants (Supplementary Fig. 10a) but there are some slight variations. Thr61, for example, which forms a direct hydrogen bond to the characteristic $O_2$ of xanthine can be replaced by serine in enzymes from other plants.

We were unable to detect XMP in seedling extracts but estimate that its concentration must be at least three orders of magnitude smaller than the GMP concentration. The high specificity of XMPP together with a $K_M$ in the low micromolar range and a high $k_{cat}$ are therefore of physiological importance, because a less specific enzyme would constantly dephosphorylate other more abundant NMPs. Additionally, a precise adjustment of the XMPP amount is apparently required because even a moderate overexpression can strongly disturb GTP homeostasis (Supplementary Fig. 6). It will be interesting to investigate how this enzyme at the crossroad between nucleotide biosynthesis and degradation is regulated and coordinated with the activity of GMP synthetase to adjust the flux of XMP towards the guanylates or into catabolism.

## Methods

**Plant material and cultivation.** *Arabidopsis thaliana* ecotype Columbia-0 was used throughout. The T-DNA mutants *xmpp-1* (SALK067037) and *xmpp-2* (SALK131244, At2g32150), *nsh1* (SALK083120, At2g36310)[11], and *gsda* (GK432D08, At5g28050)[6] were obtained from the SALK[18] and GABI-Kat[19] collections. Seedlings of *xmpp-1* and *xmpp-2* were tested for the presence of intact transcript using the primers N188 and N233 (Supplementary Table 3). The presence of *Actin2* (At3g18780) transcript was tested with the primers 1033 and 1034. The mutants *xmpp-1*, *nsh1*, and *gsda* were used to generate double and triple mutants by crossing. Complementation lines of *xmpp gsda* and *xmpp nsh1 gsda* were generated by transformation with construct H453 (Supplementary Fig. 3c,d).

Arabidopsis and *N. benthamiana* were cultivated under long-day conditions (16 h light of 85 μmol m$^{-2}$ s$^{-1}$, 22 °C/8 h dark, 20 °C) at 60% relative humidity. For metabolite analysis of Arabidopsis seeds, six individual plants of each genotype were grown next to each other in a randomized setup. Xanthosine, guanosine and guanine were quantified directly after seed harvest. Nucleotide analysis was performed with six-month-old seeds. Seedlings were grown on half-strength Murashige and Skoog medium. For metabolite analysis, each plate contained all genotypes representing one biological replicate. To equalize germination, plates were stored in the dark at 4 °C for 2 days before transfer to the growth chamber. Five days after germination half of the seedlings were transferred into the dark while the other half was kept under long-day conditions. After 48 h, whole seedlings were harvested. For the characterization of the complementation lines, seedlings were grown on separate plates for 10 days after germination and placed into the dark for 2 days.

**Cloning.** The coding sequence of *XMPP* was amplified from cDNA obtained from flower RNA introducing *Nco*I and *Xma*I sites with the primers N188 and N189 or *Cla*I and *Xma*I sites with the primers N359 and N360. Via *Cla*I and *Xma*I, the coding sequence was introduced into pXCS-HAStrep[20] (vector V13, construct X130) to generate a C-terminal hemeaglutinin (HA)- and Strep-tagged protein by transient expression in *N. benthamiana*. Similarly, it was introduced into pXCS-YFP[21] (V36, X131) for producing a C-terminal yellow fluorescent protein (YFP)-tagged protein. The coding sequence of YFP in V36 was exchanged for a mTFP1 coding sequence amplified from a wave_1T vector[22] with the primers P243 and P244 introducing *Xma*I and *Sac*I sites. This generated a vector for the expression of a cytosolic cyan fluorescent protein (V108). For production of an N-terminal Strep-tagged protein, the coding sequence was cloned into pXNS2cpmv-strep[5] (V90, construct X144) via *Nco*I and *Xma*I. To obtain the complementation lines *xmpp gsda::C* and *xmpp nsh1 gsda::C*, we cloned a genomic fragment of XMPP including 999 bp upstream of the translation start codon with an introduced coding sequence for a TwinStrep tag for N-terminal tagging (Supplementary Fig. 3d). For this, pXNS2pat-Strep-sl (V28) was generated by cloning the annealed primers 1709 and 1710 into the binary vector pAMPAT-MCS (accession number AY436765) opened with *Xho*I and *Eco*RI. V28 was used to generate pXNS2patTwinStrep-sI (V163) by opening V28 with *Kas*I and *Nco*I and introducing a second Strep tag with a preceding linker sequence formed by the annealed primers P1097 and P1098. *XMPP* genomic DNA was amplified using the primers N188 and N189 and cloned into V163 via *Nco*I and *Xma*I. Into this construct the *XMPP* promoter, amplified with the primers P1094 and P1103, was cloned via *Asc*I and *Xho*I replacing a 35 S promoter cassette (construct H453). To generate untagged Arabidopsis XMPP in *Escherichia coli* strain BL21 (DE3) for antibody production, *XMPP* amplified with the primers N188 and N189 was cloned into pET30nco-CTH[23] (V48) using *Cla*I and *Sma*I for the *XMPP* cDNA and *Cla*I and *Eco*RV for the vector (H325).

**Phylogentic analysis.** Sequences were obtained from Phytozome V12.1 (Supplementary Table 1) using the protein sequence of XMPP from *Arabidopsis thaliana* as query. A multiple alignment (Supplementary Data 1) was generated with muscle hosted at the website of the European Bioinformatics Institute. The Maximum Likelihood method and the JTT matrix-based model[24] were employed to infer the evolutionary history within the MEGA X[25] software. Bootstrapping with 1000 repeats was performed and the results are shown at the respective branches. To start the heuristic search, the Neighbor-Join and BioNJ algorithms were applied to a matrix of pairwise distances estimated using the JTT model, and the tree topology with the best log likelihood was selected. The evolutionary rate differences among sites was modeled with a discrete Gamma (G) distribution of five categories (G = 1.1995). In all, 8.28% of sites were treated as invariable. In total, 76 amino acid sequences were analyzed excluding positions with more than 5% gaps. The final dataset comprised 234 positions. Evolutionary analyses were conducted in MEGA X[25].

**Subcellular localization.** *Nicotiana benthamiana* leaves were used for transient Agrobacterium (*Rhizobium radiobacter*)-mediated (co-)expression of the constructs X131 for production of XMPP-YFP and V108 for production of mTFP1 as a cyan fluorescent cytosolic marker. Abaxial leaf surfaces were analyzed using a Leica TSC SP8 microscope equipped with an HC PL APO CS2 ×40 1.10 water immersion objective (Leica, Germany). To prevent crosstalk, images were obtained by sequential scanning with an excitation of 448 nm for mTFP1 (emission 465–495 nm) and 514 nm for YFP (emission 524–539 nm). Images were processed using the Leica Application Suite Advanced Fluorescence software (Leica Microsystems).

**Protein purification and determination of kinetic constants.** Transient expression of Strep-tagged proteins from constructs X130 and X144 were performed by infiltrating *N. benthamiana* leaves with *Rhizobium radiobacter* at an optical density of 0.5 (at 600 nm). Three days after infiltration 0.75 g of leaves were ground in 1.5 mL buffer E containing 100 mM HEPES (pH 8.0), 100 mM NaCl, 5 mM EDTA (pH 8.0), 0.005% Triton X-100, 10 mM dithiothreitol, 1:625 diluted Biolock (IBA Life Sciences), 1:10 diluted protease inhibitor (complete protease inhibitor cocktail, Roche). After centrifugation, 40 μL of StrepTactin Macroprep (IBA Life Sciences) was added to the supernatant and incubated for 10 min on a rotation wheel at 4 °C. The matrix was washed three times with 1 mL buffer W1 (100 mM HEPES (pH 8.0), 100 mM NaCl, 0.5 mM EDTA (pH 8.0), 0.005% Triton X-100, 2 mM dithiothreitol) and centrifuged at 700 g for 30 s between washes. Then the matrix was washed twice with 1 mL buffer W2 (5 mM HEPES (pH 8.0), 100 mM NaCl, 0.5 mM EDTA (pH 8.0), 0.005% Triton X-100, 2 mM dithiothreitol). For batch elution 35 μL of buffer W2 containing 2.5 mM biotin was used. Elution was repeated and fractions were pooled[5,26]. Phosphatase activity was measured at 22 °C with the Enzchek Phosphate Assay Kit (Thermo Fisher Scientific) according to the manufacturer's instructions but in a total volume of 200 μL. In brief, 2-amino-6-mercapto-7-methyl-purine (MESG) and inorganic phosphate produced from the XMPP-dependent reaction serve as two substrates for purine nucleoside phosphorylase, resulting in a product exhibiting an absorbance at 360 nm. For kinetic measurements, XMPP amounts were adjusted to obtain linear initial rates at all XMP concentrations. Protein concentrations were determined with bovine serum albumin standards in a Coomassie-stained sodium dodecyl sulfate (SDS) gel using an Odyssey Fc Imager (LI-COR Biosciences). To determine the kinetic constants, the data were plotted and fitted with the GraphPad Prism 4 software. For the substrate screen and for testing whether XMPP is inhibited by guanosine or xanthosine, XMPP with a C-terminal HAStrep tag (expressed from X130) was used.

**Crystallization and structure determination.** Crystallization of XMPP was successful only after 13 C-terminal residues had been removed, likely because the region is disordered as predicted by the Phyre2 server[27] and the XtalPred server[28]. The corresponding cDNA (encoding Met1 to Ser250) was amplified with the primers X0001 and X0002 using a codon-optimized synthetic full-length DNA as a template (Supplementary Table 3). The product was cloned via the In-Fusion method using *Nde*I and *Xho*I into the pET41b expression vector containing a coding sequence for a C-terminal His tag and the construct was transformed into *E. coli* BL21 (DE3). Protein expression was induced after the cells had reached an optical density of 0.6–0.8 (at 600 nm) by the addition of 0.5 mM IPTG followed by continued culturing for 14–16 h at 20 °C. Collected cells were sonicated and centrifuged in buffer A (50 mM Tris, pH 8.0; 100 mM NaCl) and 1 mM MgCl$_2$. The protein was purified using a HisTrap HP column with buffer A and eluted using buffer A plus 500 mM imidazole and subsequently subjected to size exclusion chromatography using a Superdex-200 column equilibrated with buffer A plus 5% (v/v) glycerol. For solving the structure in complex with XMP, a catalytically inactive Asp12 to Ala mutant called XMPP(D12A) was purified the same way.

Purified enzymes in buffer A with 5% (v/v) glycerol were concentrated to 8–15 mg mL$^{-1}$ and subjected to crystallization using the sitting drop vapor diffusion method at 22 °C. Crystals of XMPP in its ligand-free form were produced using a crystallization solution of 0.1 M Tris (pH 8.5) and 20% (w/v) PEG6000. For XMPP(D12A) complexed with XMP, crystals which had been grown in solution containing 0.1 M HEPES (pH 7.0) and 18% (w/v) PEG12000 were further soaked for 1 h in crystallization solution plus 5 mM XMP, 5 mM MgCl$_2$, and 20% (v/v) glycerol. Data collection was performed at 100 K on beamline 7 A at the Pohang Accelerator Laboratory (Korea) with a 0.5° oscillation angle and 20% (v/v) glycerol as cryoprotectant (Supplementary Table 2). Collected data were processed using the HKL2000 software[29] and the high-resolution cutoff was based on a CC$_{1/2}$ statistical value of approximately 0.6 (refs. [30–32]). The space group of both crystals was P2$_1$, with one monomer in the asymmetric unit. The structure of unliganded XMPP was solved by molecular replacement using the Phaser utility of PHENIX software[33]. A search model was generated by the Phyre Ensembler[34] and Sculptor utilities[35]. Specifically, the structures of the phosphatase Sdt1p from *Saccharomyces cerevisiae* (PDB: 3NUQ, 3OPX, and 3ONN) were used for the generation and truncation of a search model. The initial electron density map calculated based on a solution from the Phaser utility did not produce a high-quality map, even with 1.4-Å resolution. Extensive manual model building was required using the program COOT[36] and subsequent refinement using PHENIX[33] produced a highly reliable electron density map. During refinement, we observed unequivocal electron density for a metal ion on the $F_o - F_c$ map. Subsequently, the structure of XMPP(D12A) in complex with XMP was determined using the refined structure of unliganded XMPP as a starting model. We clearly identified electron density for XMP in the vicinity of the metal-binding site of XMPP(D12A). Details of data collection and refinement are listed in Supplementary Table 2.

**Functional analysis of XMPP variants.** As for the enzyme purified from plants, the Enzchek Phosphate Assay Kit (Thermo Fisher Scientific) was used for the activity assays of enzyme purified from *E. coli*.

C-terminal His-tagged full-length XMPP and various mutants were expressed and purified by affinity chromatography using a HisTrap HP column with buffer A and were eluted using buffer A containing 500 mM imidazole. A Hi-Prep 26/10 column equilibrated with buffer A was further used as final step for desalting and buffer exchange. Genes for mutant XMPPs were constructed by *Dpn*I-mediated

site-directed mutagenesis using mutagenic primers (Supplementary Table 3). Enzymes mutated in Asp14, His78, His82, Thr111, Asn112, and Lys159, respectively, caused problems in expression or solubility and were excluded from further analyses.

Activity and steady-state kinetic analyses were performed according to the manufacturer's instructions, but at 30 °C. We validated that all components in the reaction mixture were present at saturating concentrations and were not limiting the measured rates. In the activity assay, 18 nM XMPP was added per reaction containing 50 mM Tris (pH 7.5), 200 μM MESG, 1.25 mM MgCl₂, and 1 U of purine nucleoside phosphorylase. A higher XMPP concentration of 145 nM was employed for the relatively inactive variants D12A, D183A, and D184A. Activities were determined after 140 s of reaction with 125 μM XMP. For the steady-state kinetic analysis, the reaction was initiated by adding the indicated concentrations of XMP and the initial linear rate determined for the first 35 s. The $K_M$ and $V_{max}$ values were obtained using the SigmaPlot software (Systat Software).

**Electrophoresis and immunoblotting.** Proteins were extracted with buffer E (1:2 tissue to buffer ratio) and 15 μL of supernatants were separated by electrophoresis on a 12% SDS-gel and transferred by semi-dry blotting to a nitrocellulose membrane (0.45 μm pore size, Thermo Fisher Scientific). The membrane was blocked with 5% milk powder in TBS-T (20 mM TRIS-HCl, pH 7.6; 150 mM NaCl; 0.1% (v/v) Tween 20) and was incubated with custom-made rabbit polyclonal anti-XMPP antibody (1 μg mL⁻¹) in TBS-T with 0.5% (w/v) milk powder over night at 4 °C. The membrane was washed three times for 10 min with TBS-T and incubated with secondary goat anti-rabbit IgG horseradish peroxidase conjugated antibody (RABHRP1, 1:5000, Sigma-Aldrich) in TBS-T for 1 h at room temperature and washed as before. For detection, SuperSignal West Femto Maximum Sensitivity Substrate (Thermo Fisher Scientific) was used. The chemiluminescence was detected by a Lumi-Imager F1 (Hoffmann-La Roche). For detection of YFP, membranes were incubated with monoclonal anti-GFP antibody from mouse (Roche 11814460001; clones 7.1 and 13.1; 1:15000 diluted). This antibody does not bind to mTFP1. Anti-mouse IgG conjugated to alkaline phosphatase (Sigma-Aldrich A3562; 1:10,000 diluted) was used as a secondary antibody.

**Production of custom-made polyclonal anti-XMPP antibody.** Untagged Arabidopsis XMPP protein was produced from construct H325 in *E. coli* strain BL21(DE3) and precipitated in inclusion bodies. Cells were grown in 0.5 L of Luria-Bertani medium to an optical density of 0.5 (at 600 nm) and then induced by 0.5 mM isopropyl-β-D-thiogalactoside. After 3 h of induction, cells were harvested, the pellet was resuspended in 27 mL of lysis buffer (50 mM Tris-HCl (pH 8.0), 0.25% (w/v) sucrose, 1 mM EDTA (pH 8.0), 7.5 mg mL⁻¹ lysozyme), vortexed, and incubated on ice for 30 min. Cells were disrupted by sonication on ice and 70 mL of detergent buffer (20 mM Tris-HCl (pH 7.5), 2 mM EDTA (pH 8.0), 200 mM NaCl, 1% (w/v) deoxycholic acid, and 1% (v/v) Nonidet P-40) was added. The lysate was centrifuged at 5000 *g* for 10 min, the supernatant was removed, and the pellet was resuspended in 80 mL washing buffer (0.5% (v/v) Triton X-100 and 1 mM EDTA (pH 8.0)). Centrifugation and resuspension were repeated until a tight pellet was obtained. The pellet was washed in 80 mL of 70% ethanol (v/v), resuspended in 1 mL of freshly prepared PBS by sonication, and used for commercial antisera production in rabbit and antibody purification (immunoGlobe Antikörpertechnik GmbH).

**Metabolite analysis.** The extraction method was adapted from Hauck et al.[16]. For metabolite analysis, frozen plant material was homogenized with steel beads using a mixer mill MM 400 (Retsch). Tube racks were pre-cooled in liquid nitrogen. Ten milligrams of seeds were ground with one 4 mm steel bead and five 2 mm steel beads for 4.5 min at 28 s⁻¹. For seedlings, 50 mg were ground with five 2 mm steel beads for 4.5 min at 28 s⁻¹. For the extraction, 10 mM ammonium acetate buffer (pH 7.5) pre-warmed to 60 °C was used and samples were immediately transferred to a heat block at 95 °C for shaking at 1000 rpm for 10 min. For seeds and seedlings, 1 mL and 500 μL of extraction buffer were used, respectively. Following the heat treatment, samples were chilled on ice for 5 min. After centrifugation at 20,000 *g* for 10 min at 4 °C, 80% of the supernatant was transferred to a new tube to remove beads and cell debris. By centrifugation at 45,000 *g* for 15 min at 4 °C particles which might block the HPLC system were removed.

Xanthosine, xanthine, guanosine, guanine, urate and allantoate were quantified using an Agilent HPLC 1200 system with a Polaris 5 C18-A 50 ×4.6 mm column (Agilent Technologies) coupled to an Agilent 6460 C series triple quadrupole mass spectrometer. Except for urate, measurements were performed in the positive ion mode. Ammonium acetate (10 mM, pH 7.5) and 100% methanol served as solvents A and B to form the following gradient: 0 min, 5% B; 1.5 min, 5% B; 3.5 min, 15% B; 6 min, 100% B; 7 min, 100% B; 7.1 min, 5% B; and 13 min, 5% B[5]. The flow rate was 0.8 mL min⁻¹ and samples of 20 μL were injected. Seed extracts were diluted 50 fold for the quantification of xanthosine, guanosine and guanine. Seedling extracts were not diluted for the quantification of xanthosine, urate and allantoate and 100-fold diluted for the quantification of guanosine. External standard dilutions spiked 1:10 into the wild-type matrix were used for quantification of xanthosine, guanosine and guanine. For the quantification of urate and allantoate, the *xmpp gsda* matrix was used. The MS parameters are listed in Supplementary Table 4. Measurements failing the quality criteria for retention time, qualifier to

quantifier ratio or with a signal to noise ratio below 10 were called "not detected". The quantification of nucleotides was performed as recently described[14,37].

**Statistical analysis.** Data for the determination of kinetic constants were plotted and analyzed using the GraphPad Prism 4 software. For metabolite data, the R software (version 1.2.5042) and CRAN packages multcomp and sandwich were used to perform two-sided Tukey's pairwise comparisons and to consider heteroscedasticity of the dataset with the sandwich variance estimator[38,39]. Using the same packages, two-sided Dunnett's comparisons to the reference group (XMPP + 10 μM XMP) were performed for the inhibitor test. The number of replicates, the test values and the multiplicity-adjusted *p* values are reported in the Source Data file.

**Accession numbers.** Sequence data from this article can be found with the following locus identifiers: *XMPP*, At2g32150; *NSH1*, At2g36310; *GSDA*, At5g28050, *Actin2* At3g18780.

**Reporting summary.** Further information on research design is available in the Nature Research Reporting Summary linked to this article.

## Data availability

All data generated or analyzed during this study are included in this published article (and its Supplementary Information file). Mass spectrometry raw data can be supplied upon request. Atomic coordinates and structure factors of XMPP with and without XMP bound have been deposited in the Protein Data Bank (PDB) under the accession codes 7EF7 and 7EF6, respectively. Source data are provided with this paper.

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

## Acknowledgements

We thank André Specht and Hildegard Thölke for technical assistance and Anting Zhu for generating the clones X130, X131, and X144; Christel Schmiechen for generating the clone H453 and Mingjia Chen for generating the vector V108 as well as Ludwig Hothorn for advice concerning the statistical analysis. This work was supported by the Deutsche Forschungsgemeinschaft (DFG) grants WI3411/8-1, INST 187/741-1 FUGG, and GRK1798 "Signaling at the Plant-Soil Interface" for C.-P.W., and by the National Research Foundation of Korea (NRF) grant 2020R1A4A1018890 by the Korea government (MSIT) for S.R.

## Author contributions

C.-P.W. devised the project and supervised the work except for the generation of the crystal structure and the biochemical analysis of enzyme mutants, which was supervised by S.R. C.-P.W., S.R., K.J.H., and S.-Y.Y. designed the experiments and interpreted the results. K.J.H. performed the biochemical analyses of XMPP produced in plants, generated the Arabidopsis mutants, and performed the metabolite analyses of nucleosides and purine catabolic products. H.S. performed the metabolite analysis of nucleotides. N.M.-E. performed the confocal microscopy and the characterization of the T-DNA lines. C.-P.W., M.V.-H., and M.H. made bioinformatic analyses and M.V.-H. performed preliminary assays leading to the discovery of XMPP. C.-P.W. performed the phylogenetic analysis. S.-Y.Y. determined the crystal structure and made the biochemical analyses of the XMPP variants. K.J.H. conducted most of the statistical analyses. K.J.H., C.-P.W., and S.-Y.Y. prepared the figures. C.-P.W. and S.R. wrote the manuscript with the contribution of K.J.H. and S.-Y.Y. The manuscript was revised by all authors.

## Funding

## Competing interests

The authors declare no competing interests.
