## [Peer Review File · Nature Communications]

Initiation of cytosolic Plant Purine Catabolism involves a monospecific Xanthosine Monophosphate PhosphataseReviewers' Comments:

Reviewer #1:

Remarks to the Author:

This is thorough and solid study that identifies a highly specific XMP phosphatase in plants and demonstrates its role in purine catabolism in vivo. The authors also present the atomic crystal structure of the recombinant enzyme free and bound to XMP to explain the high affinity for this nucleotide, and validate the role of the residues identified in the active site by site-directed mutagenesis and enzymatic assays.

The article is nicely written, the results clearly described and the conclusions well supported. The findings are of high relevance for the field of nucleotide metabolism.

This reviewer has no criticism or suggestions to improve the manuscript further.

Santiago Ramón-Maiques

Reviewer #2:

Remarks to the Author:

The manuscript entitled « Initiation of Plant Purine Catabolism involves a monospecific Xanthosine Monophosphate Phosphatase” by Heinemann and co-workers reports the identification of a XMP-specific phosphatase (XMPP) and provides information on purine catabolism from adenylylate and guanylylate nucleotides. The authors also present a crystal structure of XMPP.

Based on sequence homology with yeast nucleotidases, the authors identified five proteins potentially having nucleotide monophosphate hydrolase activity and focused on the three most conserved ones. They expressed and purified the corresponding Arabidopsis tagged proteins and found highly specific XMP phosphatase activity in vitro for one of the constructs but no GMP phosphatase activity. Using single, double and triple mutants they found some correlation between the lack of XMPP and XMP accumulation in vivo. Finally, the authors present a crystal structure of XMPP that gives clues on the specificity of the enzyme. The experiments are well performed and overall support (although sometime rather weakly, see below) the conclusions.

These findings, and notably the high specificity of XMP, are potentially interesting and could reveal some important metabolic specificities of plants by contrast to microorganisms or animals but as such the work presented in the manuscript does not allow to draw strong conclusions on why this enzyme should be so specific and how it affects purine metabolism as a whole.

Major points:

1- The in vivo role of XMPP is indirectly deduced from double and triple mutants with *gsda* and *nsh1*. Due to the remaining GMPS activity in these mutants, these results are not fully conclusive. It would have been much more direct to combine the *xmpp* mutant with a *gmpp* mutant and measure purine derivatives in the double and single mutants. As such the conclusions are uncertain and weak. The choice of the *gsda* and *nsh1* mutants rather than a *gmpp* mutant is not explained.

2- The phenotypical consequences of the lack of XMPP are not established by the authors. If XMPP is an important enzyme, the lack of XMPP, either alone or combined, should have important consequences on purine metabolism, cell and/or organism physiology. Could XMP be toxic? Is there an impact of purine recycling on proliferation or survival? There is no evidence that this could be the case. As such, the in vivo results appear with limited general interest to me.

Dear Dr. Pattison,

Please find below the response to your and the reviewers' comments

COMMENT: "We do not wish to be prescriptive in how you address the concerns of reviewer #2. We do not necessarily consider the conditional nature of the phenotype (i.e. that the most appreciable differences in nucleotide metabolism are seen in xmpp gsdA as opposed to xmpp single mutants) as a barrier to publication. However, we do feel that further work to determine when XMPP might be relevant in wild type plants (with a functional GSDA) would strengthen the case for publication. "

REPLY: We fully understand this concern and have obviously made every effort to find differences between the wild type and the XMPP mutants. At this point, we do not have any additional data in hand that would further elucidate the functional / physiological role of XMPP in plants. In this work our achievements are (1) the discovery of the enzyme, (2) its placement into the plant metabolic network based on robust in vivo data and (3) a thorough biochemical characterization.

This work is a significant contribution to the discovery of gene function, which is a very laborious effort following the massive gene discovery of recent years. When browsing the annotation of the Arabidopsis genome, one realizes that we are far from understanding the in-depth function of most genes – including many of those whose knockouts create phenotypes.

From the long list of genes with unknown in vivo function in plants, we eliminated one in this work, although admittedly it will require considerable further research to understand its exact physiological role – and this is unfortunately far beyond the scope of our report.

COMMENT: "2- The phenotypical consequences of the lack of XMPP are not established by the authors. If XMPP is an important enzyme, the lack of XMPP, either alone or combined, should have important consequences on purine metabolism, cell and/or organism physiology. Could XMP be toxic? Is there an impact of purine recycling on proliferation or survival? There is no evidence that this could be the case. As such, the in vivo results appear with limited general interest to me."

REPLY: We show that the lack of XMPP reduces the direct flow of adenylates into purine catabolism. We also show how this is hardwired in the context of maintaining guanylate homeostasis. So yes, both processes have drastic "consequences on purine metabolism" as manifested by depletion of purine nucleotide catabolite pools and loss of control over guanylate pools in the respective mutants.

Could XMP be toxic? XMP concentrations in the plant are so low that one needs state-of-the-art sample preparation and MS technology to detect it at all. Although concentrations are slightly higher in XMPP mutants, they are still very low. This data and the lack of a macroscopic phenotype, lets us believe that the primary role of XMPP is not to avoid XMP toxicity. A manipulation of XMP amounts from outside to test toxicity is not straightforward, because Arabidopsis does not take up mononucleotides from growth media. Nucleosides are taken up, but xanthosine cannot be phosphorylated to XMP in the plant. It has been clearly demonstrated that the only metabolic fate of xanthosine in Arabidopsis is its degradation by purine catabolism.

It is long known that a lack of purine catabolism does not have strong phenotypic consequences on macroscopic level (“impact of purine recycling on proliferation or survival”, Witte und Herde, 2020, PMID 31641078). It has been suggested that the process contributes to nitrogen recycling from the purine ring but despite some efforts to show this in the last decade (e.g. Soltabeyeva et al. 2018, PMID 30190419) fully convincing evidence that also excludes toxicity effects from metabolites accumulating in mutants, is lacking so far – in our opinion. This may be so, because protein recycling contributes far more nitrogen, masking the contribution of purine base catabolism to the overall nitrogen budget.

COMMENT: Furthermore, we appreciate that two *xmpp* T-DNA mutant alleles and a complementation line for the *xmpp gsdA* double mutant (*xmpp gsdA::C*) have been generated. However, we note that Figure 3a does not include measurements for the complementation line. Given that the lack of xanthosine accumulation in *xmpp gsdA* is perhaps the most compelling evidence of XMPP activity in planta, we would request that this experiment is repeated with the complementation line included (to rule out the possibility that metabolite levels are altered by second site mutations).

REPLY: We have repeated the dark-exposure part of the experiment, where the difference between the *GSDA* mutant and the *XMPP GSDA* mutant was most apparent. The wild type, the *GSDA* mutant, the *XMPP GSDA* double mutant and the complementation line of this double mutant were included. The data is presented in the new Supplementary Figure 7, and shows that the effect of the *XMPP* mutation in *gsda* background is reversed by the introduction of an *XMPP* transgene. It is thus clear, that the mutation of *XMPP* and not any second site mutation causes the metabolic alterations.

COMMENT: 1- The in vivo role of XMPP is indirectly deduced from double and triple mutants with *gsda* and *nsh1*. Due to the remaining GMPS activity in these mutants, these results are not fully conclusive. It would have been much more direct to combine the *xmpp* mutant with a *gmps* mutant and measure purine derivatives in the double and single mutants. As such the conclusions are uncertain and weak. The choice of the *gsda* and *nsh1* mutants rather than a *gmps* mutant is not explained.

REPLY: In all organisms, GMPS is an essential enzyme for the synthesis of guanylates. It is therefore impossible to work with *GMPS* null-mutants since the mutation is lethal. A knockdown mutant may be viable in Arabidopsis – although this has never been shown -, but one can expect drastic effects on guanylate and in consequence overall nucleotide metabolite pools. Surely these plants would also show macroscopic phenotypes. In the context of such a drastic scenario, it would not have been easier to show the function of XMPP in vivo, in our opinion.

Reviewers' Comments:

Reviewer #2:

Remarks to the Author:

Following the reply by the authors to point #1 of reviewer #2

GMPS is not an essential gene in all organisms. It has been deleted from the budding yeast genome and the resulting mutant strain is perfectly viable, provided that guanylic precursors are added to the growth medium (guanine auxotrophy)(Dujardin et al. Gene 1994 Feb 11;139(1):127-32. doi: 10.1016/0378-1119(94)90535-5).

In fact, this possible bypass of GMPS by guanine or guanosine was represented by the authors themselves on the right part of figure 1a.

Reviewer #3:

Remarks to the Author:

The manuscript "Initiation of Plant Purine Catabolism involves a monospecific Xanthosine Monophosphate Phosphatase" by Heinemann et al. represents a solid piece of work as already highlighted in the two first round reviews. All results presented follow a clear line aiming to describe all facets of a newly identified gene/protein from phylogenetic analysis, protein biochemistry, analysis of Arabidopsis mutants (single, double, triple, complementation) up to the crystal structure. I am not aware of any other example where a formerly black box was filled with such comprehensive information.

In my opinion, work on nucleotide metabolism is a underestimated research topic in the plant field, in contrast to the situation e.g. in mammals. However, recent work already shows that nucleotide metabolism not only serves to provide building blocks for nucleic acid synthesis, but is heavily involved in regulation of growth and acclimation processes. Therefore, I regard this work of high general interest.

The XMPP knock-out mutants have clear molecular phenotypes, but these are not resembled in growth or developmental phenotypes. I understand that this can be an issue. However, in their response letter, the authors claim to have made "every effort" to identify such phenotypes and when looking at other publications from the same group it is clear that all tools necessary for proper phenotyping of plant mutants are at hand and will have been used. Currently, lack of phenotype is not discussed in the manuscript, but I assume that not few readers will search for corresponding information. I suggest to clearly address this point and to state that no growth phenotype was observed and maybe even show this in a subfigure with seedlings with visible roots (XMPP is highly expressed in root tissues according to the "Genevisible" expression database) from the WT, XMPP knockouts and complementation lines. Experiments performed in this context could be briefly summarized and explanations or even speculations why blocking of purine catabolism especially at the initiation level does not produce phenotypes could be provided.

Reviewer #2 (Remarks to the Author):

Following the reply by the authors to point #1 of reviewer #2

GMPS is not an essential gene in all organisms. It has been deleted from the budding yeast genome and the resulting mutant strain is perfectly viable, provided that guanylic precursors are added to the growth medium (guanine auxotrophy)(Dujardin et al. Gene 1994 Feb 11;139(1):127-32. doi: 10.1016/0378-1119(94)90535-5).

In fact, this possible bypass of GMPS by guanine or guanosine was represented by the authors themselves on the right part of figure 1a.

REPLY: The reviewer is right, that technically *GMPS* is non-essential in yeast, because the resulting auxotrophy upon *GMPS* mutation can be healed by adding guanine to the growth media. Still, I would argue that biologically *GMPS* is an essential gene even for yeast, because in its natural habitat there will be hardly enough guanine to survive without it. For a complex multicellular organism like *Arabidopsis thaliana*, *GMPS* is surely essential – technically and biologically. It would be impossible to rely on guanine or guanosine salvage alone since these metabolites would have to reach every cell in sufficient amounts and salvage metabolism would have to be active enough in every cell to supply enough GMP. It is highly unlikely that this is the case and even if it were possible, such a manipulation would not make the elucidation of the XMPP function in plants easier, but rather complicate things because it creates a highly artificial metabolic situation where the system relies on salvage while we want to investigate degradation.

Reviewer #3 (Remarks to the Author):

The manuscript “Initiation of Plant Purine Catabolism involves a monospecific Xanthosine Monophosphate Phosphatase” by Heinemann et al. represents a solid piece of work as already highlighted in the two first round reviews. All results presented follow a clear line aiming to describe all facets of a newly identified gene/protein from phylogenetic analysis, protein biochemistry, analysis of *Arabidopsis* mutants (single, double, triple, complementation) up to the crystal structure. I am not aware of any other example where a formerly black box was filled with such comprehensive information. In my opinion, work on nucleotide metabolism is a underestimated research topic in the plant field, in contrast to the situation e.g. in mammals. However, recent work already shows that nucleotide metabolism not only serves to provide building blocks for nucleic acid synthesis, but is heavily involved in regulation of growth and acclimation processes. Therefore, I regard this work of high general interest.

The XMPP knock-out mutants have clear molecular phenotypes, but these are not resembled in growth or developmental phenotypes. I understand that this can be an issue. However, in their response letter, the authors claim to have made “every effort” to identify such phenotypes and when looking at other publications from the same group it is clear that all tools necessary for proper phenotyping of plant mutants are at hand and will have been used. Currently, lack of phenotype is not discussed in the manuscript, but I assume that not few readers will search for corresponding information. I suggest to

clearly address this point and to state that no growth phenotype was observed and maybe even show this in a subfigure with seedlings with visible roots (XMPP is highly expressed in root tissues according to the “Genevisible” expression database) from the WT, XMPP knockouts and complementation lines. Experiments performed in this context could be briefly summarized and explanations or even speculations why blocking of purine catabolism especially at the initiation level does not produce phenotypes could be provided.

REPLY: XMPP is a highly expressed gene in every tissue of *A. thaliana* (Klepikova et al., 2016, Plant Journal 88, 1058-1070; http://bar.utoronto.ca/efp/cgi-bin/efpWeb.cgi?dataSource=Klepikova_Atlas). Therefore phenotypes might be expected in many tissues in the respective mutants not only in roots. We have included the new Supplementary Fig. 9 showing that the *XMPP* mutant does not show any apparent phenotypes. We have also added an additional paragraph at the end of the section “The function of XMPP in purine nucleotide catabolism in vivo” briefly discussing this finding.